# The Role of Taurine in Skeletal Muscle Functioning and Its Potential as a Supportive Treatment for Duchenne Muscular Dystrophy

**DOI:** 10.3390/metabo12020193

**Published:** 2022-02-19

**Authors:** Caroline Merckx, Boel De Paepe

**Affiliations:** 1Department of Neurology, Ghent University Hospital, 9000 Ghent, Belgium; boel.depaepe@ugent.be; 2Neuromuscular Reference Center, Ghent University Hospital, 9000 Ghent, Belgium

**Keywords:** taurine, osmolytes, muscle, Duchenne Muscular Dystrophy, mdx

## Abstract

Taurine (2-aminoethanesulfonic acid) is required for ensuring proper muscle functioning. Knockout of the taurine transporter in mice results in low taurine concentrations in the muscle and associates with myofiber necrosis and diminished exercise capacity. Interestingly, regulation of taurine and its transporter is altered in the mdx mouse, a model for Duchenne Muscular Dystrophy (DMD). DMD is a genetic disorder characterized by progressive muscle degeneration and weakness due to the absence of dystrophin from the muscle membrane, causing destabilization and contraction-induced muscle cell damage. This review explores the physiological role of taurine in skeletal muscle and the consequences of a disturbed balance in DMD. Its potential as a supportive treatment for DMD is also discussed. In addition to genetic correction, that is currently under development as a curative treatment, taurine supplementation has the potential to reduce muscle inflammation and improve muscle strength in patients.

## 1. Introduction

Taurine or 2-aminoethane-sulfonic acid is primarily a free occurring sulfur-containing amino acid. Unlike most other amino acids, it is not a building block for proteins, yet classifies as a conditionally essential amino acid that is abundant in excitable tissues such as brain, retina, heart, and skeletal muscle, where intracellular concentrations range from 20 to 70 mmol/kg. Taurine is either taken up from diet, for example from fish and meat, or can be synthesized from other amino acids such as cysteine or methionine. Taurine has versatile functions: it plays an important role in osmoregulation, acts as a stabilizer of the cell membrane and of proteins, has anti-oxidant and anti-inflammatory functions, regulates mitochondrial tRNA activities, is involved in calcium homeostasis, etc. [1,2,3].

In this review, we focused on the role of taurine in muscle disease, especially in Duchenne Muscular Dystrophy (DMD), a progressive muscle wasting disorder affecting approximately 1 per 5000 male births [4]. Muscle weakness is conspicuous in the hip- and pelvic area first, and later spreads to distal regions. Patients become wheelchair-dependent in their early teens and eventually require cardiac and respiratory care since the muscles of the heart and respiratory system are affected in a life-threatening manner. While awaiting curative treatments to enter the clinic, glucocorticoids are the standard of care, and can prolong life-expectancy of DMD patients [5,6]. Although the precise mechanism by which glucocorticoids slow down disease progression in DMD is not completely understood, its anti-inflammatory action might play a crucial role. However, the use of glucocorticoids negatively influences bone health, which is already impaired in DMD patients [7]. A comprehensive overview of emerging genetic therapies in DMD is provided in the review of Sun et al. [8]

The genetic cause of DMD is mutations in the dystrophin gene, located on chromosome X, which hampers the production of functional dystrophin protein. The latter is a key-component of the dystrophin-associated protein complex (DAPC) that provides stability to muscle fibers during contraction and relaxation by connecting the intracellular actin cytoskeleton to the basal lamina [7]. Besides membrane stabilization, DAPC also fulfils a role in signal transduction. Dystrophic muscles encounter chronic inflammation, oxidative stress, and ischemia. Eventually these detrimental processes lead to loss of muscle mass and muscle fibrosis [7].

This review explores the physiological role of taurine in skeletal muscle and focuses on the consequences of its disturbed balance in DMD. The therapeutic potential of taurine as a dietary supplement for DMD will be scrutinized.

## 2. Involvement of Taurine in Physiological Skeletal Muscle Functioning

### 2.1. Knowledge Gained from Knockout Models

A first clue towards an important role for taurine in the muscle was its relatively high abundance. Proper insights regarding the function of taurine in physiological muscle function were acquired upon the generation of taurine transporter (TauT) knock-out (KO) mouse models and ablation of muscle taurine content. Lack of taurine impaired the conductance velocity of the muscle without affecting nerve conductance speed [2]. In addition, exercise performance was seriously hampered in TauT KO mice as shown by a significantly lower running speed. In a different experimental set-up, the total running distance was only 20% of the distance travelled by age-matched wild type (WT) mice [2,9,10]. Besides running tests, the reduced exercise capacity of TauT KO mice also became apparent during a weight-loaded swimming test that showed an 80% decrease in swimming time compared to WT [11]. This study reported structural changes in morphology of TauT KO muscle; however, Warskulat et al. hypothesized that hampered exercise performance was likely attributed to muscle dysfunction, resulting from taurine deficiency. In evidence, serum creatine kinase levels are increased in TauT KO and serum lactate levels were raised after exercise [2]. Some of the pathological characteristics of TauT KO models such as necrotic myofibers and reduced exercise capacity resemble the features of the mdx mouse model [12,13]. Another approach to evaluate the effect of taurine depletion is the use of guanodinoethane sulfonate (GES). GES, a taurine transporter antagonist, reduces muscle taurine content by 60% [14]. Previously taurine was shown to enhance calcium uptake and release in myofibers concurrently with an increase in force production, whereas myofibers derived from GES-treated mice showed reduced force production at relevant stimulation frequencies. Interestingly, fatigue was reportedly attenuated upon GES-treatment [14].

Furthermore, the lifespan of TauT KO male mice is significantly lower than those of WT mice, 511 and 686 days, respectively. Reduced life expectancy together with increased expression of p16INK4a, an indicator of senescence, in TauT KO mice allowed to hypothesize that taurine might be involved in aging [15,16]. Furthermore, it was suggested that taurine delays muscle-specific senescence including sarcopenia in tumor necrosis factor (TNF)-α stimulated L6 myogenic rat cells. In evidence, differences in the regulation of inflammation, autophagy, and apoptosis have been reported upon taurine treatment [17]. In addition, TNF-α stimulation of L6 myogenic rat cells hampered muscle differentiation which could be restored by taurine. Presumably this effect was mediated through the PI3/AKT signaling pathway since myocyte enhancer factor-2 (MEF-2), a transcription factor involved in myogenic differentiation, was markedly decreased after knockdown of AKT [17]. Furthermore, expression of TauT increased during muscle differentiation and was even further enhanced upon binding of MEF-2 to the promotor region of TauT [18].

In conclusion, depletion of muscle taurine levels either through TauT KO or via pharmacological inhibition of the taurine transporter by GES alters force output and exercise performance. Therefore, taurine seems essential for the preservation of physiological muscle function. The sections below provide an overview of the main cellular processes in which taurine plays a role in relation to the muscle.

### 2.2. Taurine and Its Role in Osmotic Homeostasis

Exposure to hyperosmolar conditions can induce several detrimental cellular effects including interference with transcriptional and translational activity, induction of oxidative stress, DNA damage, and can even elicit apoptosis under certain conditions [19,20]. Thus, safeguarding the osmotic equilibrium is essential for ensuring cellular health. When cells are exposed to an environment high on NaCl, fluid is retracted from the intracellular compartment causing cellular shrinkage, molecular crowding, and increased ionic forces. In order to counteract these deleterious processes, the cell’s response is a regulatory volume increase (RVI) that includes activation of inorganic ion transporters (e.g., Na^+^/K^+^/2Cl^−^ cotransporter, the Na^+^/H^+/−^, and the Cl^−^/HCO_3_^−^ exchanger), allowing an influx of ions accompanied by osmotic uptake of water [19,20,21,22,23]. However, this condition is unfavorable over time due to increased intracellular ionic forces that could interact with macromolecules. Thus secondarily, accumulation of organic osmolytes (e.g., taurine) that replace inorganic electrolytes results in normalization of ionic strength whilst preserving cellular volume and protein stability [19,20]. Rather than stimulation of de novo synthesis, osmotic stress is most likely to enhance cellular import of taurine [20]. Both the designated TauT as well as the proton-coupled amino-acid transporter (PAT) 1 are capable of accumulating taurine in the cell [24]. However, TauT is considered as the principal transporter of taurine in muscle cells as evidenced by a 98% reduction of taurine content in a TauT KO mouse [10]. Transcription of TauT mRNA is upregulated under hypertonic conditions due to binding of Nuclear Factor of Activated T-cells 5 (NFAT-5) to the 5′ flank region of the TauT gene. NFAT-5, also known as tonicity responsive element binding protein (TonEBP) acts as a transcription factor of *SLC6A6*, the gene encoding TauT, and thus allows cellular accumulation of taurine [25,26].

Exercise can affect the osmotic balance in muscle fibers. Muscle subjected to an intensive exercise protocol resulted in increased myofiber volume, cross-sectional area, and water concentration by more than 15%, indicative for muscle fiber swelling [27,28]. The rise in myofiber water content can be partially explained by water production in cellular metabolic processes that take place during exercise [27,28,29]. Additionally, intracellular solute concentrations might be elevated during exercise as a result of phosphocreatine splitting and increased lactate and H^+^, ensuing water influx in order to retain osmotic balance [28] and could contribute to myofiber swelling as well. This volume increase is followed by a compensatory mechanism named the regulatory volume decrease (RVD) that releases electrolytes (e.g., K^+^, HCO_3_^−^, Cl^−^) and osmolytes such as taurine concurrently with water in order to normalize cellular volume [19,20,26,27,28,29,30,31]. Thus, taurine is released in order to counteract myofiber swelling, a phenomenon which occurs during exercise [28,31].

### 2.3. Taurine and Its Role in Protein and Membrane Stabilization

The stabilizing effect of taurine is mentioned in many papers. However, the mechanism by which taurine is able to exert stabilization is poorly described. In order to comprehend this characteristic, it is important to understand its interaction with water molecules, considering the chemical properties related to its molecular structure. One of the most popular hypotheses that could explain protein stabilization by osmolytes is based on preferential exclusion [32,33,34,35]. This principle builds on unfavorable interactions between proteins and osmolytes in terms of Gibbs adsorption isotherm [32]. In a denatured state, the area of the peptide backbone by which osmolytes can interact is larger and results in increased Gibbs energy (unfavorable). In order to reduce these interactions, the thermodynamic component drives the folding equilibrium towards its native state, also referred to as the osmophobic effect, which is associated with a much lower Gibbs energy. This simplified explanation implies that in the presence of stabilizing osmolytes, the Gibbs energy of the denatured state is much higher compared to Gibbs energy of the folded state [32]. Therefore, the folded protein conformation is favored and osmolytes act as protein stabilizers [32,33]. In general, the presence of osmolytes results in a specific distribution of water molecules around the proteins in a preferential hydrated state and osmolyte exclusion from the protein backbone [32,33,34,35]. 

Furthermore, stabilizing actions have been attributed to taurine, as well as direct interaction with protein side chains [33,34,35]. The amino group of taurine orients itself preferentially to the protein side chain. This strengthens the hydrogen bonded network of water surrounding the protein and stabilizes its native form. The latter appears to contradict the preferential exclusion theory; however, such interactions between osmolytes and side chains have also been discussed by Bolen et al. [32]. It should be noted that protein side chains are associated with other characteristics than the protein backbone and favorable interactions between osmolyte and side chains might occur. Supposedly, the latter does not substantially alter the protein folding state [32]. In the article by Brudziak et al., the protein was hydrated in the presence of taurine [35]. This might suggest that besides limited interactions between osmolytes and protein side chains, the protein is still preferentially surrounded by water molecules. Taurine was able to increase the thermal stability of both lysozyme and ubiquitin protein [35,36,37,38], although the extent of stabilization was protein specific [35].

In addition to protein stabilization, membrane stabilizing properties of taurine were hypothesized by Huxtable and Bressler [39]. Taurine inhibits the activity of phospholipid methyltransferase, which catalyzes the methylation of phosphatidylethanolamine to form phosphatidylcholine and thus taurine could alter the composition and consequently the properties and stability of phospholipid membranes [40,41,42]. In evidence, the presence of taurine decreased the viscosity of erythrocytic membranes, suggesting taurine might increase membrane fluidity [43]. 

Eccentric muscle contraction might induce denaturation of myofibrillar proteins as hypothesized by Paulsen et al. [44]. In addition, the unfolded protein response (UPR) is activated during exercise [45,46] which might indicate that proteins struggle to maintain native folding conformations. Interestingly, prolonged exercise increased the denaturation temperature of albumin, pointing to enhanced thermal stability [47]. The importance of taurine in protein stabilization is illustrated in the TauT KO mouse model [15]. It is assumed that a lack of taurine allows accumulation of unfolded and/or misfolded proteins in skeletal muscle which activates expression of genes involved in the UPR. Thus, taurine plays a key role in protein homeostasis of skeletal muscle [15,48].

### 2.4. Taurine and Its Role in Oxidative Stress

Under physiological conditions, reactive oxygen species (ROS) are balanced by antioxidant mechanisms that detoxify reactive species. A limited amount of ROS is produced during exercise and exerts advantageous effects on force generation. In addition, low levels of ROS might protect against injury through adaptations in cellular signaling upon regular training exercise [27,47,49], whereas high levels of ROS are associated with muscle dysfunction [50]. Although direct scavenging of the main ROS (e.g., superoxide anion (O_2^−^_), hydrogen peroxide (H_2_O_2_), and hydroxyl radical (·OH)) by taurine is considered unlikely [51,52], taurine is believed to relieve oxidative stress through neutralization of hypochlorous acid (HOCl) and upregulation of antioxidant enzymes [53,54,55]. Upon inflammation, neutrophils become activated and release myeloperoxidase (MPO). MPO catalyzes the reaction of chloride and H_2_O_2_, a classic ROS molecule, resulting in the formation of HOCl. HOCl has toxic effects and interferes with cellular processes including molecular transport and pump capacity [56]. Recently, it has been hypothesized that HOCl could alter excitation-contraction (E-C) coupling and impairs muscle force production [57]. HOCl is converted to TauCl after interaction with taurine. TauCl possesses anti-inflammatory properties and increases antioxidant enzymes including heme oxygenase 1 in murine microglial cells and muscle cells [58,59]. Taurine supplementation increased activity of antioxidant enzymes such as superoxide dismutase and catalase, measured in the blood of patients with type II diabetes [60]. Similar results were obtained in the liver and kidney of an ethanol-induced oxidative stress mouse model [61]. 

Furthermore, antioxidant effects of taurine on superoxide production and lipid peroxidation were observed in the muscle of eccentric exercised rats [62]. Similarly oxidative lipid damage was reduced by taurine treatment in a mouse model of muscle overuse [63].

### 2.5. The Role of Taurine in Mitochondrial Protein Synthesis

Taurine participates in the synthesis of mitochondrial proteins, more specifically proteins that require tRNA^(Leu)^ and tRNA^(Lys)^ with uridine on a Wobble position [64]. If the first base of the tRNA anticodon is uridine, then the classic Watson–Crick rules are substituted by Wobble base pairing rules. According to this hypothesis, H-bonds are formed between the first base of the anticodon (tRNA) and the third position of the mRNA codon. Contrastingly to Watson–Crick base pairing, Wobble pairing suggests that uridine on a Wobble position at the anticodon can form H-bonds with A, U, G, and C on the third position of the mRNA codon mRNA. Whereas, taurine-conjugated uridine tRNA will promote the formation of H-bonds between uridine and A or G on the third codon position and is required for appropriate translation to leucine (UUA/UUG) [64]. Taurine modification of uridine is required in some mitochondrial tRNAs to ensure a more specific codon-anticodon interaction and proper mitochondrial protein synthesis. Cytochrome b, ND5, and ND6 are mitochondrial proteins containing UUG codons, and synthesis of these proteins is potentially hampered if taurine conjugation of mitochondrial tRNA^Leu(UUR)^ is deficient. ND5 and ND6 are functional subunits of oxidative phosphorylation complex I, that catalyzes electron transport from NADH to coenzyme Q [64,65]. As the process of oxidative phosphorylation is one of the main sources of ROS [50] in myofibers, preservation of mitochondrial function is considered essential in the safeguarding of oxidative stress.

Specific mutations in tRNA^(Lys)^ and tRNA^(Leu)^ are associated with respectively myoclonic epilepsy with ragged red fibers (MERRF) and mitochondrial encephalopathy, lactic acidosis, and stroke-like episodes (MELAS). MERRF and MELAS are mitochondrial diseases that show disturbed protein synthesis and pathological characteristics such as exercise intolerance, thus of which some aspects resemble the TauT KO phenotype [9,64,66]. Of note, taurine supplementation in MELAS patients improved the occurrence of stroke-like episodes [67].

### 2.6. Taurine and Its Role in Calcium Homeostasis

Calcium is an essential cellular building block and a universal carrier of biological information. As a diffusible intracellular second messenger, calcium is involved in signaling transduction pathways and can regulate many different processes ranging from neurotransmitter release, bone formation, and blood coagulation to muscle contraction [68]. More specifically, calcium is essential during the E-C coupling mechanism that transforms the electrical input (action potential) to a mechanical output (contraction). Upon adequate stimulation, an action potential is generated and propagates over the sarcolemma to the transverse tubules that contain L-type calcium channels e.g., dihydropyridine receptors (DHPR). In the skeletal muscle, DHPR are in close contact with the ryanodine receptors (RyR) of the sarcoplasmic reticulum (SR), which releases calcium into the cytosol. Binding of calcium to troponin C induces the translocation of tropomyosin, thereby allowing the formation of cross-bridges between actin and myosin filaments and subsequent contraction [69].

Temporal and spatial changes of calcium concentration in cytoplasm or organelles are monitored by a multitude of calcium sensing proteins that determine the character and duration of the cellular response. The SR is an organelle involved in storage of calcium which is released into the cytoplasmic environment upon muscle stimulation. Taurine partially preserved SR function upon exposure to phospholipase C, which is known to hamper calcium transport, in SR derived from rat skeletal muscle. Furthermore, in the presence of 15 mM taurine, uptake of calcium oxalate by the SR was increased by more than 20% [39].

Similarly, taurine significantly increased the accumulation of calcium in SR of skinned EDL rat myofibers [70]. In this experiment, the membrane of the myofibers was mechanically removed which allowed to control intracellular taurine concentrations. The enhanced calcium SR load and subsequent release upon stimulation might explain the increment in force response upon depolarization. The authors hypothesized that taurine modulates SR calcium pump activity, allowing increased calcium accumulation [70]. These results are consistent with observations in human skeletal muscle fibers type I and II; interestingly, these different muscle fibers contain a different type of SR calcium transport ATPase (SERCA)-isoform, thus taurine might modulate either activity and/or calcium affinity [71]. Of note, a small decline in calcium sensitivity was observed in the presence of taurine in SR of skinned EDL myofibers [70].

Furthermore, voltage clamp experiments carried out in cardiomyocyte derived from guinea pig showed that the effect of taurine on calcium influx is highly dependent of the extracellular calcium concentration. In the presence of taurine (20 mM), calcium influx was slightly increased upon low extracellular calcium concentration (0.8 mM) and vice versa, the calcium influx was slightly yet significant decreased when extracellular calcium concentrations were high (3.6 mM). No effect of taurine on inward calcium current could be observed at an extracellular calcium level of 1.8 mM, suggesting that taurine aims at maintaining intracellular calcium homeostasis [72]. However, taurine modulated resting membrane potential regardless of extracellular calcium channels.

## 3. Pathophysiological Characteristics of Taurine in DMD

### 3.1. Regulation in Dystrophin Deficiency

Previously, it has been suggested that alterations in taurine and/or its regulation associate with dystrophinopathology [73,74,75]. Lack of dystrophin results in destabilization of DAPC and thereby rendering the sarcolemma more susceptible to contraction-induced damage. Membrane fragility is observed in dystrophin deficient muscle as evidenced by increased permeability to dyes such as Evans Blue and Procion Orange, accumulation of serum proteins such as albumin and IgG, increased levels of creatine kinase in the blood, and sarcolemma rupture in myofibers [7,76,77].

Taurine regulation is altered in dystrophic tissues of mouse models. Some studies reported a significant downregulation of the TauT in muscles of the mdx mouse model [73], whereas others reported no significant differences between mdx mice and age-matched control mice [78,79]. Seemingly, TauT is significantly downregulated in young mdx mouse and normalizes to control levels over time.

Similar to the expression of TauT, differences in muscle taurine content with age were reported as well, yet with considerable differences between studies. Taurine content of mdx muscle was comparable to controls at age 3–4 weeks, but increased by age 10 weeks [78,80]. A significant downregulation of taurine in mdx mice aged 4 and 6 weeks compared to age-matched control mice was reported [59,73], whereas other studies found no significant difference in taurine content between control and mdx mice at age 6 weeks [73,81]. A decline in taurine content was on the other hand observed in the plantaris muscle of 6-month-old mdx mice compared to wet weight yet did not hold up when compared to dry weight [79]. Contrasting results have been reported regarding the taurine content in mdx mice, however most of these studies conclude that taurine is differentially regulated in mdx mice compared to control mice at a certain time point. The expression of TauT and taurine content in the muscle of mdx mice compared to age-matched control mice is summarized in Table 1. Interestingly, muscle taurine levels increased in glucocorticoid treated mdx mice, pointing to its reestablishment by immunosuppressive therapeutic interventions [75].

As opposed to the observations in the mdx mouse model a significant increase in muscle taurine and its transporter were discerned in 8-month-old Golden Retriever Muscular dystrophy (GRMD) canine model [74]. Similarly, taurine regulation in DMD patients differs from healthy control patients, as shown by increased levels of muscle TauT protein in DMD patients [82,83]. Thus, expression of TauT is differently regulated in the mdx mouse than in the GRMD-model and DMD patients. Interestingly, the phenotype in the mdx mouse model is milder than in GRMD dogs and DMD patients. Myofiber necrosis and muscle weakness become conspicuous approximately 3 weeks after mdx mice are born. Myofiber necrosis in the mdx mouse is most explicit in the young to juvenile period, followed by necrosis at a slower pace in adult mice [74,80]. The course of disease in mdx mice thereby differs from the more progressive pathology in GRMD dogs and DMD patients, that is characterized by fatty replacement and fibrosis at an early age [74]. These differences in pathological features might be linked to regulation of TauT.

### 3.2. Role in Oxidative Stress Management and Mitochondrial Protein Synthesis

Mitochondria participate in cellular energy production by synthesizing adenosine triphosphate (ATP) through oxidative phosphorylation [84]. This process includes transfer of electrons, required for ATP-synthesis, and is inevitably linked to production of ROS. Under physiological conditions, basal levels of ROS are generated as a by-product of oxidative phosphorylation. However, ROS production is enhanced upon mitochondrial dysfunction [84]. Upon oxidative phosphorylation, electron transfer is facilitated by mitochondrial protein complexes that resides within the inner mitochondrial membrane [84]. However, Onopiuk found significantly reduced protein levels of complex III, cytochrome-c reductase, and complex V, the ATP synthase, in immortalized myoblasts of mdx mice. Of note, myoblasts of mdx mice (SC-5), and control myoblasts (IMO) were used [85]. Neither of these myoblasts expressed dystrophin protein, only dystrophin mRNA was present in control myoblasts. In addition, Onopiuk et al. showed an increase in mitochondrial membrane potential [85]. Taken together, these results suggest mitochondrial dysfunction and inevitably, increased ROS production in dystrophin deficient cells. Taurine reduces ROS by upregulation of antioxidant enzymes and might preserve electron transport chain activity by safeguarding mitochondrial protein synthesis of subunits involved in the respiratory chain. In evidence, reduced taurine levels hamper expression of ND6, subunit of complex I in the mitochondria of cardiomyocytes derived from rat [64,86].

### 3.3. Dysregulation of Calcium Homeostasis

Excessive calcium levels are observed in dystrophic myofibers; however, calcium entry through sarcolemmal tears is not considered as the main contributor to calcium overload in dystrophic myofibers. Apparently, the open probability of calcium leak channels in the proximity of micro-tears are increased, thereby allowing an increased calcium influx [87]. Additionally, increased expression and activation of store-operated calcium channels, presumably induced by calcium-independent phospholipase A2, is observed in dystrophin deficient myofibers and could contribute to calcium overload as well [84,88,89,90]. Moreover, a correlation between the dystrophic phenotype and expression of a stretch-activated channel, transient receptor potential canonical (TRPC) channel 1, was discovered in different muscles of the mdx mouse. The diaphragm of mdx mice was affected the most, as shown by increased Evans Blue permeability, and showed a significant upregulation of TRPC1 expression compared to controls [88]. Thus, the involvement of the TRPC channels might also contribute to the calcium overload as evidenced by increased expression of various TRPC channels in mdx mice. The cytoplasmic calcium concentration is not only determined by calcium channels/exchangers but also by the SR uptake and release of calcium, which plays an essential role during E-C coupling. RyR, responsible for the release of calcium from SR upon depolarization, is hyper nitrosylated in dystrophic muscle and consecutive depletion of calstabin-1 upon RyR-nitrosylation results in calcium leakage [84,89,90,91,92]. During relaxation, cytoplasmic calcium ions are sequestered by SERCA, thereby lowering its cytoplasmic concentration. In the mdx mouse model, the removal of calcium ions by the SR is hampered, suggesting reduced SERCA functioning. In conclusion, calcium homeostasis is disturbed in dystrophic muscle on many levels. Chronic cytoplasmic calcium overload will induce activation of degrading pathways mediated by phospholipase 2 and proteases which eventually can lead to myofiber death [7,90].

In addition, cytoplasmic calcium overload can induce accumulation of calcium in the mitochondria and mitochondrial dysfunction. Multiple pathways have been proposed by which mitochondrial calcium overload can induce ROS production, such as mitochondrial permeability transition (MPT)-mediated release of anti-oxidative enzymes, dislocation of mitochondrial proteins involved in electron transports including cytochrome C, induction of NO, etc. [84,93]. Mitochondrial calcium overload can elicit MPT pore formation. A permeable pore is formed that spans inner and outer mitochondrial membranes and results in mitochondrial swelling [84]. Furthermore, dystrophic muscle cells show an increased susceptibility to calcium and therefore are more prone to MPT pore formation, that eventually cause mitochondrial swelling and death [87,91]. In evidence, mitochondrial swelling, indicative for MPT pore formation, was induced at a lower calcium load in mitochondria compared to WT [84,90,92,93,94]. Interestingly, taurine is able to attenuate calcium-induced swelling of mitochondria derived from skeletal muscle [95].

## 4. Taurine Supplementation as a Therapeutic Strategy for DMD

### 4.1. Effect of Taurine on Muscle Force

A beneficial effect of taurine on muscle force remains controversial: supported by some studies yet disproved by others. A significant increase in peak twitch force was obtained in taurine supplemented (2.5% *w*/*v* in drinking water) mdx mice compared to untreated mdx mice at 4 weeks. However, this effect was abrogated in juvenile and adult mdx mice, respectively aged 10 weeks (2.5% *w*/*v*) and 6 months (3% *w*/*v*) [78,79]. Similarly, a >50% increase in maximum isometric tetanic specific force (sPo) was obtained by taurine supplementation in mdx mice aged 4 and 6 weeks [78,81]. However, no treatment effect (1 g/kg/day) on specific tetanic force was detected in muscle of mice aged 5–7 weeks, whereas fore limb force, assessed by means of a grip strength meter, was ameliorated [96]. Furthermore, a study performed by Barker reported no effects of taurine treatment (2.5% *w*/*v*) on peak twitch force, maximum specific force, nor fatigue or recovery in mdx mice treated from week 2 until week 4 [97]. Similarly, no effect on maximum specific force in 6-week-old mdx mice was observed when high doses of taurine (16 g/kg/day) were administered [98].

In 6-month-old mdx mice, taurine could not ameliorate specific maximum isometric force production. However, after a fatigue protocol consisting of recurrent electrical stimulation, the EDL of taurine treated mdx animals was more resistant to fatigue, as shown by a significant higher force production at the end of stimulation relative to force production at the beginning. Furthermore, EDL muscle of taurine treated mdx animals showed a significantly better recovery capacity at 10–60 min after stimulation than untreated mdx animals [79]. Taurine treatment in exercised mdx mice significantly increased in vivo forelimb muscle strength normalized to body weight [96,99], but did not alter locomotor activity of mdx mice. A similar finding was reported in a study that supplemented unexercised mdx mice. Taurine supplementation (±4 g/kg/day) in 6-week-old unexercised mdx mice significantly increased grip strength [81]. Whereas no effect of high taurine treatment (16 g/kg/day) was observed on normalized grip strength [98].

In summary, benefit of taurine supplementation on muscle force differed considerably between published studies. Although multiple studies have been executed, these used different concentrations, ingestion methods, and treatment protocols, which might explain the obtained conflicting results, and which hampers interpretation.

### 4.2. Effect of Taurine on Oxidative Stress and Inflammation

Although different treatment conditions and read-outs were used, multiple studies have shown anti-inflammatory and anti-oxidative effects of taurine treatment in mdx mice [59,80,81,95]. The anti-oxidant and anti-inflammatory actions of taurine have been proposed as a mechanism by which taurine protects dystrophic tissue from damage. In evidence, muscle of taurine-treated mdx mice showed significantly less NF-κβ positive fibers, TNF-α levels, neutrophil elastase, and MPO activity [59,80,81,96]. Accordingly, taurine-treated mdx mice showed lower levels of disulfide and protein thiol oxidation in muscle compared to untreated mdx mice, which could point towards protection against oxidative stress [59,81,98,100]. Similarly, ROS levels were significantly reduced upon taurine treatment in muscles of exercised dystrophic mice, as shown by dihydroethidium (DHE) staining, which reacts with O_2^−^_ [96]. In addition, taurine normalized the resting macroscopic ionic conductance (gm), a measure for ROS production, to WT levels [96].

### 4.3. Effects of Taurine on E-C Coupling

In mdx mice, the threshold of the membrane potential at which a contraction is elicited is shifted towards a more negative value than in control mice. Thus, contraction is induced upon a lower depolarization state than in WT animals [96,101]. This mechanical threshold is indicative for E-C functioning and suggests alterations in E-C-coupling and/or calcium homeostasis [96,101]. Interestingly, taurine treatment in mdx mice has been shown to partially restore the mechanical threshold [96,99,101]. Taurine treatment in mdx did not alter expression of proteins involved in E-C coupling such as calcium sensitive receptors (RyR), calcium channels (DHPR channels), calcium ATP-ase pump (SERCA) and calcium binding proteins (calsequestrin) [78]. Contrastingly, taurine supplementation (2.5% *w*/*v*) in rats significantly increased expression of calsequestrin 1 in the muscle [102]. Although in the mdx mouse and rat study the same dose of taurine was used (2.5% *w*/*v*), this discrepancy in outcome might be explained by age-dependent calsequestrin regulation, changes in calsequestrin regulation upon dystrophin deficiency, species-dependent regulation, or differences in treatment protocol. However, a specific explanation cannot be pinpointed with the current literature studies available.

### 4.4. Effect of Taurine on Histopathological Characteristics of the Mdx Mouse

Histopathological characteristics such as necrotic myofibers and fibers with centralized nuclei, indicative for regeneration ensuing myofiber damage, are conspicuous in Hematoxylin-Eosin stained sections of mdx muscles and are used to evaluate therapeutic effectiveness [59]. In the study of Barker, taurine (2.5% *w*/*v*) reduced the amount of non-contractile area in young mdx mice, whereas no effect on the percentage regenerative fibers could be observed [78]. Other studies have reported a significant increase in healthy myofibers with peripheral nuclei upon taurine treatment [59], a decrease of histopathological features and myofiber necrosis [80,96]. Thus, taurine seems to alleviate histological features related to dystrophinopathy.

### 4.5. Combinatory Use of Taurine and Glucocorticoids

Combined use of taurine (1 g/kg) and α-methylprednisolone (PDN) (1 mg/kg), a synthetic adrenocortical hormone, in the exercised mdx mouse model significantly improved muscle strength in such a way that a synergistic effect was proposed by Cozzoli et al. [103]. The increase in fore limb muscle strength after 4 weeks of treatment was significantly higher in mice that received combined therapy compared to mice treated with PDN. Similarly, the increment in muscle force normalized to body weight was remarkably elevated in mice that received taurine + PDN compared to single-drug treatments. Furthermore, combined treatment normalized the rheobase potential to WT values, however similar results were observed in taurine-treated mice. No synergistic effect of combination treatment compared to PDN-treatment was detected regarding histopathological markers that included the amount of centronucleated fibers, necrosis and non-muscle tissue [103].

Contrastingly, the study of Barker et al. reported no effects of combined treatment on muscle strength. These opposing findings might be explained by differences in experimental set-up since in the latter study therapeutic intervention was initiated more closely to onset of damage, higher taurine concentrations were used for a shorter period of time and analysis occurred at the peak of damage [97]. Besides possible synergistic effects of combined treatment, taurine could also counteract side-effects related to corticosteroids. Dexamethasone causes muscle atrophy and significantly lowers the myotube diameter, which is restored upon taurine treatment [18]. Furthermore, taurine protects against glucocorticoid-induced mitochondrial dysfunction of the bone and might attenuate corticoid-induced osteoporosis and or osteonecrosis, pathological features that are conspicuous in glucocorticoid-treated DMD patients [104,105,106].

### 4.6. Potential Caveats of Taurine Treatment

In some rat studies, different test regimes of taurine supplementation (3% *w*/*v* for 4 weeks [107], 100 mg/kg for 2 weeks [108]; 500 mg/kg for 2 weeks [109]) resulted in increased muscle taurine content, whilst other long-term studies (1% *w*/*v* ≈ 50 mg/kg for 22 weeks [110]) could not report increased muscle taurine levels. Similarly, in mouse studies muscle taurine levels were elevated upon taurine supplementation (4% *w*/*v* for 3 weeks [59]; 3% *w*/*v* for 4 weeks [79]). Interestingly, one study showed that continuous taurine supplementation (2.5% *w*/*v*) in mdx mice increased muscle taurine concentration up to the age of 4 weeks, but when treatment was prolonged up to the age of 10 weeks, muscle taurine concentration was significantly decreased in taurine supplemented mdx mice compared to untreated mdx mice [78]. Similarly, high doses of taurine (8% *w*/*v* ≈ 16 g/kg/day) added to the drinking water up until the age of 6 weeks, did not increase muscle taurine content [98]. Therefore, it might be hypothesized that the muscle taurine content is strictly regulated. Since chronic treatment might not be able to increase intramuscular taurine levels it is not known if long-term taurine treatment could effectively attenuate muscle damage. Furthermore, taurine intake (≈5 g/day) for a period of 7 days was reported not to alter muscle taurine levels in humans [111].

In general, taurine is well tolerated and safe if used in appropriate concentrations [112]. One study has reported gastro-intestinal complaints; however, taurine was used in combination with other nutritional supplements [112,113]. Another study aimed to investigate taurine supplementation in patients with end-stage renal failure. In healthy subjects, excessive taurine is excreted; however, due to kidney failure the taurine overload was not immediately cleared and these patients suffered from dizziness and vertigo [114] which was the reason to discontinue the study.

High taurine treatment (16 g/kg) in 6 week-old animals significantly reduced the body weight of mdx mice by more than 20% and shortened tibia length by 10%. Of note, the body weight and tibia length of untreated mdx mice were comparable to those of WT animals in this experiment [98]. Similarly, taurine treatment (3% *w*/*v*) for 4 weeks in adult mdx mice (5 months old) reduced body weight and muscle mass of mdx mice; however, the weight of these animals still exceeded that of WT animals [79]. Interestingly, no effect of taurine on body weight was observed in WT animals. Obesity is commonly observed in DMD patients, especially in glucocorticoid-treated patients [7,115,116]. Therefore, it remains unclear if taurine-induced decreased body weight would be disadvantageous in these patients. Obviously if growth is hampered, this should be avoided. A study conducted in piglets reported a reduced gain to feed ratio upon higher taurine supplementation (3% taurine diet) compared to untreated piglets whilst supplementation with 0.3% taurine might improve growth [117]. Terrill et al. have hypothesized that high taurine supplementation in young animals interferes with the taurine synthesis pathway, shifting the equilibrium to the left inducing increased cysteine levels in the plasma which could hamper growth [98].

### 4.7. Other Nutritional Supplements Used in Duchenne Muscular Dystrophy

Approximately 50–65% of DMD patients are using vitamins or nutritional supplements [115,116], which indicates that the quest for supportive treatment, including but not limited to taurine, in DMD is relevant. A detailed overview of nutritional supplements that are under investigation for the treatment of DMD is provided in the review of Boccanegra et al. [118].

The current nutritional guidelines for patients are very similar to those used for the general population. If serum 25-hydroxy-vitamin D drops below 30 ng/mL or calcium intake is low, the use of vitamin D and, respectively, calcium intake under the form of calcium citrate or calcium carbonate is advised in order to stimulate bone health, which could be impaired in DMD patients [116,117,119]. In addition, other nutrients are currently under investigation. Coenzyme Q10, a naturally occurring anti-oxidant, significantly improved muscle strength in steroid-treated DMD patients [118,120]. Similarly, beneficial effects of L-arginine, creatine, omega-3 fatty acids have been observed in clinical trials [118].

## 5. Conclusions

Taurine is involved in numerous processes such as protein stabilization and osmotic homeostasis and appears to be indispensable for physiological muscle function as evidenced in the TauT KO mouse. We further zoomed in on DMD, a severe muscle disorder that showed altered regulation of taurine and its transporter. Multiple facets of dystrophinopathology and potential mechanisms on how taurine might act on these pathological features have been proposed and discussed in view of the mdx mouse model. Promising results have been obtained in the mdx mouse model in terms of inflammation, muscle strength, oxidative stress, etc. and led to the conclusion that taurine supplementation is relevant for DMD pathology. Former clinical trials conducted to evaluate the anti-aging or mood stabilizing effects of taurine have shown that taurine is well tolerated and considered safe upon appropriate use. However, up until now, no clinical trials have been conducted that evaluated taurine as a treatment for DMD patients. We propose that taurine has the potential to act as supportive therapy in combination with glucocorticoids for the treatment of DMD patients and further studies should be conducted to evaluate the effectiveness of chronic taurine supplementation.

## Figures and Tables

**Table 1 metabolites-12-00193-t001:** Summary of TauT and taurine expression in the mdx mouse.

Timepoint of Analysis	Muscle TauT Content	Muscle Taurine Content	Muscle Type	Reference
18 days	↓ in mdx mice	≈ controls	quadriceps	73
22 days	/	≈ controls	quadriceps	80
28 days 28 days	≈ controls ≈ controls	↓ in mdx mice ≈ controls	quadriceps tibialis anterior	73 78
6 weeks 6 weeks 6 weeks	↓ in mdx mice / /	≈ controls ↓ in mdx mice ≈ controls	quadriceps quad/gas tibialis anterior	73 59 81
10 weeks	≈ controls	↑ in mdx	tibialis anterior	78
6 months	≈ controls	↓ in mdx mice (wet weight) ≈ controls (dry weight)	EDL (TauT); plantaris (taurine)	79

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
