# Peer review of "The Role of Taurine in Skeletal Muscle Functioning and Its Potential as a Supportive Treatment for Duchenne Muscular Dystrophy"

_metabolites, 2022, doi:10.3390/metabo12020193_

Round 1
Reviewer 1 Report
In the review article, Drs Merckx and Paepe introduce the roles of taurine on the skeletal muscle functions, its dynamics in Duchenne muscular dystrophy and the effectiveness of its treatment on the diseases. Authors review a lot of information about pathology, physiology, biochemistry, and etc. about the skeletal muscles, however, there are many explanations of basic knowledges as literature levels especially in sessions 2 and 3. Authors need to brush up the explanation of basic knowledges and more focus on the evidence of taurine.
Authors introduce many evidence and studies about taurine, particularly in its physiological roles, however, most of them are obtained not only from skeletal muscles but various other tissues and cells. It is unclear whether these identified roles of taurine can be adapted to skeletal muscle. So, authors need to explain which tissue the evidence was obtained from and whether it is the same in skeletal muscle.
It is also advisable to add figures and schemes to make the readers easier to understand.
Specific comments:
There are coexisted expressions about taurine transporter knockout including TauT KO, TauT-/-, and TauT deficient, and about canticum and Ca2+. Please, make them uniform expressions.
Once an abbreviation has been set, the word must use the abbreviation. For example, “taurine transporter” and “knock out” appeared many times after setting of their abbreviations.
Lines 67-69: In the previously published article, Warskulat et al. showed the double value of serum CK level in sedentary state and the elevated lactate release during exercise in the TauT KO mice compared to those in the respective states of age-matched WT mice, and so, suggested that the pathological abnormalities in the skeletal muscle directly induced by taurine deficiency may related to the reduced exercise performance in the treadmill experiment in the TauT KO mice. Accordingly, this previous article did not hypothesize that hampered exercise performance was more likely attributed to lack of taurine rather than structural aberrations.
Line 69: “lactate serum level” should be improved to “serum lactate level”.
Line 73: “taurine muscle content” should be improved to “muscle taurine content”.
Line 74: Please add reference following “60%”.
Line 79: Please move “respectively” following “686 days” and add “,” between them.
Lines 82-83: Please add information of “in vitro model” such as cell type and species (origins).
Line 84: Please add reference following “taurine treatment”.
Line 111: Please delete “taurine transporter”.
Lines 130: It wonders why “anoxia” is suddenly mentioned.
Line 139: Please add reference following “exclusion”.
Lines 133-163: Is the explaining about the membrane stabilization by taurine able to be also adapted to skeletal muscle cell (fiber), and are there evidence in skeletal muscles?
Line 177: Please add reference following “mouse model”.
Lines 199-200: Reference [63] was study in the liver and kidney. Does ethanol induce oxidative stress in the skeletal muscle, and does taurine have anti-oxidative effect in the ethanol treated skeletal muscle?
Lines 200-201: This sentence is unclear what types of exercise were loaded in the animals and what kinds of oxidative stress were induced by the exercises.
Lines 221-226: Are there any evidence about the effectiveness of taurine treatment on the mitochondrial encephalomyopathies?
Line 245: “SR-derived rat skeletal muscle” is unclear. Is it “SR derived from rat skeletal muscle”?
Line 254: “sarcoplasmic reticulum” should be “SR”.
Lines 258-266: The evidences were obtained from cardiomyocytes. They could be adapted to skeletal muscle?
Lines 277-279”: The sentences are repetition of the sentence in lines 133-114.
Lines 280-281: “at day 18 and 42” and “at day 28” are the number of days old?
Lines 279-284 and Lines 285-292: It is unclear what difference between these sentences of the first and second paragraphs mentioned.
Lines 294-295: Because there is a variety of evidence for increased, unchanged, or decreased taurine content in mdx mice, the meaning of this statement is unclear. Which are the “most these studies”, how is taurine differently regulated in the model mice, and when is the “certain time point”? Please, write it more carefully since this is the conclusion of these studies.
Lines 259-297: It wonders why this sentence about glucocorticoid treatment and immunosuppressive therapeutic intervention are suddenly mentioned here.
Lines 300-302: The words of “alterations” and “abnormal” are unsuitable, because authors explain in this sentence that the regulations of taurine and its transporter expression in the DMD patients are discerned similarly to canine model. Please, write clearly which “increase” or “decrease”. In addition, the evidence in the DMD patients mean the regulation of urinary taurine excretion by TauT in kidney, and therefore, it is unclear whether this evidence can mention taurine content increases in the skeletal muscle in the patients.
Line 305: Please change “becomes” to “become”.
Line 330-331: In the section 3.2., most of sentences explain the oxidative stress management and protein synthesis in the mitochondria, and only last one sentence mentions about the taurine. Please, more focus on the evidence of taurine, since this article should review about taurine. This comment is the same for Section 3.3.
Line 372: The first paragraph introduces both positive and negative effects of taurine on muscle force. So, please improve “positively”.
Line 379: “1g/kg/day” is lower than the dose of 2.5% wt/vol?
Line 380: “in vivo force” is unclear.
Line 387: Is “treated mdx animals” the same to “taurine treated mdx mice”?
Line 394: What does (±) mean, particularly in “minus”?
Lines 395: The sentence “Whereas no……observed [102]” is repetition in lines 383-384.
Lines 397-400: Based on the fact that the protocols and results are different in each study, it is unclear what the authors want to mention from those evidence.
Lines 407-410: This information is also repetition.
Lines 423-426: Reviewers recommend writing this evidence in more detail on the effect of taurine treatment on the expression of E-C coupling protein.
Lines 426-427: It is unclear why taurine treatment could increase calsequestrin expression in rats whereas the expression was unchanged in mice. Are there differences in treatment protocol of taurine (dose or duration) or species difference in the regulation of calsequestrin expression between rats and mice? If the reasons were known or unknown, please describe them.
Line 439: Please explain that PDN is the synthetic adrenocortical hormone.
Line 441: Please add reference following “Cozzoli et al”.
Line 448: Please add information what was histopathological markers.
Line 453-455: “synergistic antimicrobial effects” of taurine with dexamethasone is related to the pathology of dystrophy?
Lines 459-461: It also is wondering how the effect of taurine on glucocorticoid-induced mitochondrial dysfunction of the bone is related to the pathology of dystrophy?
Lines 472-474: This evidence from [102] is in mdx mice?
Lines 475-477: In this sentence, authors explain that taurine treatment even if in long-term could not increase muscle taurine level. It is unclear which the authors would like to mention that taurine treatment is not effective on the dystrophy or the increment of muscle taurine content is not necessary on the effective of taurine treatment on the dystrophy.
Lines 502-515: Section 4.7. is not related to taurine on dystrophy, and so, it is wondering if this section is necessary to this review of taurine on dystrophy or not.
In reference.
Authors need to check the style of reference, especially in the journal style including the large character, and bold and italic.
Reference 14: “J or Amino acid” is incorrect. “Amino Acids” is correct.
Reference 16: “Taurine” should be changed to “Adv Exp Med Biol”.
Reference 69: Please add information of issue number and pages.
References 86 and 94: Two references may be same.
Reviewer 2 Report
The review on taurine’s role in skeletal muscle health and muscular dystrophy by Caroline Merckx et al is well written and informative for muscle field especially muscular dystrophy. Most of the studies in mice using MDX model which is not well represented the DMD patient phenotype. MDX mice is larger than parental normal mice. Overall, taurine treatment improves muscle inflammation, alleviate oxidative stress and muscle pathology of DMD mice despite controversial role on muscle strength which depend on time and dose of treatment. The review summarized large amounts of literature and recommend further clinical trial of combination of taurine with corticoid is of importance. Recommend accept with minor wording edits:
Comments are as below:
- Line 55, “Lessons learned” Should be “knowledge gained or learned” since the Knock-out model reveals the positive role of taurine, not only negative role.
- Line 147, Confirmation should be Conformation?
- There are several times of use “repercussion”. The meaning is not clear to me and recommend used a commonly used term at this circumstance.
- The first sentence in the conclusion should be removed. This is a sentence should be in beginning of the review or abstract not at the end.
Reviewer 3 Report
This is a highly informative review on the role of taurine in muscle during health and disease. In particular, the Authors discuss the ways in which taurine balance is disturbed in DMD and how it impacts disease progression. Finally, the Authors review the evidence in support of a potential therapeutic intervention in DMD in the form of taurine dietary supplementation. Although generally well written, this manuscript does need to be edited for English language and grammar.
Minor comments:
Line 43-47, add reference
Line 134-149 Please add the relevant reference where appropriate, rather than adding them all together at the end of the paragraph.
Line 203-218 See above
Line 306-310, add reference
Line 314-317, add reference
Line 429-431, add reference
